# Evaluation of Transport and Location Policies to Realize the Carbon-Free Urban Society

**Shinichi Muto [1],\*, Hiroto Toyama [2] and Akina Takai [2]**

1   Graduate Faculty of Interdisciplinary Research, University of Yamanashi, 4-3-11 Takeda,
    Kofu 400-8511, Yamanashi, Japan
2   Integrated Graduate School of Medicine, Engineering and Agricultural Sciences, Faculty of Engineering,
    University of Yamanashi, 4-3-11 Takeda, Kofu 400-8511, Yamanashi, Japan; g21tc008@yamanashi.ac.jp (H.T.);
    g21tc006@yamanashi.ac.jp (A.T.)
\*   Correspondence: smutoh@yamanashi.ac.jp; Tel.: +81-55-220-8599

**Abstract:** The Japanese Government has declared that it will become carbon-free by 2050. Urban planning to realize a carbon-free society is proposed in the context of urban transport policy, which are policies to agglomerate urban facilities and link among them by public transport. However, transport and location policies to regulate land use are afraid to generate an economic loss. It is important to evaluate not only the effects of reducing GHG emissions but also economic influence. In this paper, we built the Computable General Equilibrium and Urban Economic (CGEUE) model, which modeled the transport and location behavior of each economic agent for a detailed area explicitly. We evaluated some transport and location policies such as (1) conversion from fossil fuel vehicles to electric vehicles, (2) improvement of public transport, (3) environmental tax and (4) making city compact by using the CGEUE model. As a result, it can be concluded that the combination policy of improving the public transport policy and environmental tax is the most effective under the conditions of these simulation results.

**Keywords:** carbon-free society; GHG emissions; policy economic evaluation; CGE model; CUE model

## 1. Introduction

The Japanese Government has declared that it will realize a carbon-free society by 2050 [1]. It is important to discuss what kind of policies should be introduced to realize a carbon-free society. The introduction of policies for decarbonization is likely to generate economic losses while having environmental improvement effects. Therefore, it is necessary to introduce policies that can efficiently reduce greenhouses gas (GHG) emissions while suppressing the economic impacts as possible.

In Japan, subsidy policies for developing and diffusing low-carbon vehicles or environmental taxes were introduced in order to reduce GHG emissions of automobiles at the national level. The environmental improvement effects and economic losses of the GHG emissions reduction policies by Muto et al. applying the Computable General Equilibrium (CGE) model [2,3]. On the other hand, the public transport construction policies to convert from automobiles to public transport or the compact city policies to decrease automobile mileage were considered for introduction at the urban level in order to reduce GHG emissions. Since the effects or impacts of those policies differ depending on the urban structure, it is important to be evaluated for each urban.

In this paper, we discuss desirable policies for the realization of a carbon-free society at the urban level. When conducting analysis at the urban level, it is necessary an evaluation model that incorporates both transport and location behavioral models. Furthermore, in order to suppress the economic impacts as possible, it is important to be able to evaluate the policy economically. The Land-Use Transport Interaction (LUTI) model is a typical urban model that is incorporated both transport and land-use (Wegener 2004 [4]). However, they

were not based on the microeconomic foundation, and the economic impacts of policies are not able to be evaluated by social welfare. The Computable Urban Economic (CUE) model was developed by Ueda et al. [5], and they reformulated the LUTI model based on the microeconomic foundation.

In the CUE model, the economic behavior of households and firms is modeled based on the microeconomic foundation. The transport behavior models, including route choices and location choice models, are incorporated into them. Muto et al. [6] and Zhang et al. [7] applied the CUE model and clarified the effectiveness of some policies by evaluating the effect of GHG emission reduction and economic influence. However, the CUE model was a partial equilibrium model that targeted only the equilibrium condition of the land markets.

A general equilibrium model dealing with both transport and location behaviors evolved in the field of urban economics. Thorpe [8] applied the general equilibrium model based on the Alonso type model [9], which is incorporated the transport and location behavior in an urban area and evaluated policies for reducing environmental externalities caused through transport trips from the perspective of social welfare. Unteroberdoerster [10] also applied the general equilibrium model based on the Core–Periphery model, which was built by Krugman [11] and developed by other researchers, and analyzed policies for reducing cross-border pollutions. Although these papers made great contributions to the theoretical elucidations of policies for reducing urban externalities, they are not a practical policy evaluation model because they are basically a monocentric city model and do not have the transport network in the transport behavior model.

Anas developed a practical general equilibrium urban economic model. Anas and Xu [12] proposed a general equilibrium model that is close to real society by relaxing the assumption of the monocentric city model and analyzing labor distribution, and so on. However, the model was a simple supposition for producing behavior of firms. Robson et al. [13] reviewed researches on the CGE model analysis, and he stated that the CGE model is based on the general equilibrium model and has an I–O (input–output) structure. In this sense, the model of Anas and Xu [12] was not a CGE type model.

Anas and Liu [14] then developed the RELU–TRAN (Regional Economy, Land Use and TRANsportation) model, which is incorporated an I–O structure into the general equilibrium model of Anas and Xu [12]. Robson and Dixit [15] also built a general equilibrium model that incorporated the I–O structure and dealt with both transport and location behavior. Muto et al. [16,17] also proposed an integrated model of the CGE and CUE model in consideration of the I–O structure that is based on the CUE model developed by Muto et al. [18]. However, although those CGE models clearly indicate the transport firm, they had a problem that their production behavior model was not realistic. In particular, when transport improvement was implemented, it was not considered to improve the production efficiency of a transport firm.

In this paper, we propose a CGEUE (Computable General Equilibrium and Urban Economic) model based on the integrated model of the CGE and CUE model developed by Muto et al. [16,17]. It is a general equilibrium model that incorporates the I–O structure and deals with both transport and location behavior. The difference from the previous model is that the CGEUE model carefully formulates the production behavior model of the transport firm. Specifically, we consider that the required time also affects the labor and capital input of the transport firm. Therefore, when the required time is saved by the transport improvement, the labor and capital input of the transport firm also is assumed to be reduced because the transport firm produces transport services by inputting labor and capital and moving his transport facility. The savings of labor and capital input generate benefits by spillover effects through the market, leading to decreasing production costs for transport firms and lower transport prices. This makes it possible to evaluate public transport improvement projects more realistically than ever before.

In addition, since our model has a CGE model structure, it is possible to accurately evaluate the impact of the introduction of taxes and consider the petroleum refinery products sector as one sector of the I–O structure. Therefore, we numerically evaluated the

effects of reducing GHG emissions and influences of economic activities for some policies such as electric vehicle diffusion, environmental taxes, public transport improvement and making a compact city, and clarifying which policy is more effective.

## 2. Structure of the CGEUE Model

### 2.1. Concept of Modelling

We built the CUE model to make measuring theoretical urban economic analysis numerically [18]. The CUE model is based on urban economic models or the LUTI models [19]. It is characterized by modeling based fully on a microeconomics foundation so that it can evaluate urban transport or land-use policies consistently with welfare economics or a cost–benefit approach. However, the CUE model is partial equilibrium focusing on only land markets, so it cannot evaluate indirect influences that are generated through market mechanisms. Expanding the CUE model to general equilibrium flamework remains an important task.

The CGE (Computable General Equilibrium) model is another analytical model to evaluate the public policies, which outputs equilibrium price or quantity in all markets [13]. Because the CGE model is the general equilibrium formulation, it can evaluate the whole indirect effects. However, the CGE model does not incorporate the region as divided spaces or areas. Although the SCGE (Spatial CGE) model, which incorporates the concept of space to the CGE model, was developed, the SCGE model is necessary to the interregional input–output table for the numerical simulation in principle. When we apply the SCGE model to detailed areas in urban, it is difficult to make the interregional input–output data. Therefore, we resigned the application of the SCGE model for such areas.

We attempted to build the CGEUE model that is integrated the CUE model and CGE model. The CGEUE model is the new urban economic model that is incorporated all market equilibrium of commodities or product factors, unlike the CUE model. Anas [20,21] already built such an integrated model, which is applied a discrete choice model to location behavior based on general equilibrium formulation. In contrast, we adopted the location behavior formulation by CES (Constant Elasticity of Substitution) function approach in order to maintain consistency in the CGE modeling. Additionally, we treated the markets separately by the integrated markets cleared for the whole urban area and the markets cleared for each zone in order to consider the balance of appropriate computational complexity and necessary information.

### 2.2. Assumption of the CGEUE Model

We suppose that an urban area is divided into some zones, and there are households, representative firms producing goods, a real estate firm providing land services and transport firms in each zone.

The behavior of agents is explicitly formalized as expenditure or cost-minimizing in the framework of the CGE model. The interactions at the inside of markets are modeled by the price adjusting mechanism. In regard to the markets, the prices of agriculture goods, manufacturing goods and factors are cleared in integrated markets for the whole urban area, and the ones of commercial, private services, and real estate services are cleared in markets for each zone, and the one transport service is cleared in markets for each OD (Origin and Destination). The formulation of a transport firm's behavior allows us to model based on a characteristic that the ODs are considered in transport.

### 2.3. Formulation of the CGEUE Model

#### 2.3.1. Firm's Behavior

The firm produces by inputting intermediate goods and product factors. In regards to the agriculture and manufacturing firms, the representative firm decides and produces the volume of products in the whole region, and they decide the zone where those product goods are produced. On the other hand, service firms decide the volume of product in each zone $i$ for the demand of its zone $i$ (See Figure 1).

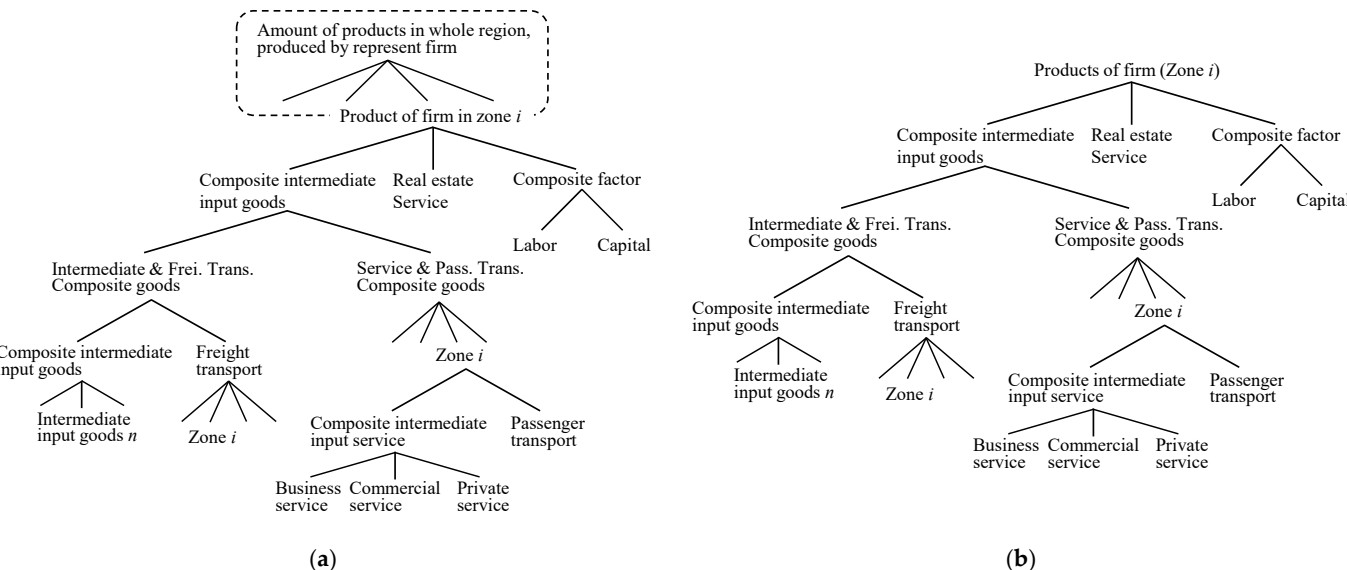

**Figure 1.** Tree structure of firms' production behavior: (**a**) agriculture or manufacturing firms; (**b**) service firms.

The behavior determining the zone's produced goods for the agriculture and manufacturing firms are formulated as below.

$$p_m y_m = \min_{y_m^i} \sum_i p_m^i y_m^i \tag{1a}$$

$$\text{s.t. } y_m = \gamma_m \left[ \sum_i \alpha_m^i \left\{ \beta_m^i y_m^i \right\}^{\frac{\sigma_m - 1}{\sigma_m}} \right]^{\frac{\sigma_m}{\sigma_m - 1}} \tag{1b}$$

where:

$y_m$, $p_m$ = producing volume of goods $m$ and price of goods $m$;
$y_m^i$, $p_m^i$ = producing volume and price of zone $i$;
$\alpha_m^i$, $\beta_m^i$ = share parameters ($\sum_m \alpha_m^i = 1$, $\sum_m \beta_m^i = 1$);
$\gamma_m$ = scale parameter;
$\sigma_m$ = elasticity of substitution.
By solving Equation (1), we obtained the product functions in each zone $i$.

$$y_m^i = \frac{1}{\gamma_m \left( \beta_m^i \right)^{1-\sigma_m}} \left( \frac{\alpha_m^i}{p_m^i} \right)^{\sigma_m} \Psi_m^{\frac{\sigma_m}{1-\sigma_m}} \cdot y_m \tag{2}$$

where:

$$\Psi_m = \sum_i \left( \alpha_m^i \right)^{\sigma_m} \left( \frac{p_m^i}{\beta_m^i} \right)^{1-\sigma_m}$$

By substituting Equation (2) into Equation (1a), we obtained the price of goods $m$.

$$p_m = \frac{1}{\gamma_m} \Psi_m^{\frac{1}{1-\sigma_m}} \tag{3}$$

Firms determine the inputting volume of intermediate goods and product factors for producing the volume of each zone $i$. These behaviors are shown in Figure 1. In the first step, a firm determines inputting volume of composite intermediate input goods, real estate service and composite product factor, respectively. In the second step, for inputting composite intermediate input goods, they determine the inputting volumes of two composite goods; intermediate goods and freight transport services and services and

passenger transport services. The freight and passenger transport services are assumed to be necessary to input the intermediate goods and some services, respectively. In the third step, for inputting composite intermediate goods and freight transport services, they decide the inputting volume of intermediate goods and freight transport services, respectively, and for intermediate goods, they decide the inputting volume of n goods. Additionally, for freight transport services, they choose the origin zone from which freight transport services generate. On the other hand, for inputting composite services of some services and passenger transport services, they choose the inputting zone and determine the inputting volume of composite services in each zone, and for inputting volume of services in each zone, they decide the inputting volume of composite intermediate input services and passenger transport services, respectively. For inputting composite intermediate input services, they decide the inputting volume of business, commercial and private services, respectively. In the last step, for composite product factors, they determine the inputting volume of labor and capital, respectively.

These firms' behaviors when producing goods are formulated by the cost-minimizing program constraint to keep product technology constant. The formulation is shown as follows.

$$p_m^i y_m^i = \min_{z_m^i, \; x_{REm}^i, cf_m^i} \left[ q_m^i z_m^i + p_{RE}^i x_{REm}^i + \left(1 + \tau_m^i\right) pf_m^i cf_m^i \right] \tag{4a}$$

$$\text{s.t.} y_m^i = \gamma_m^i \left[ \alpha_{Zm}^i \left\{ \beta_{Zm}^i z_m^i \right\}^{\frac{\sigma_m^i - 1}{\sigma_m^i}} + \alpha_{REm}^i \left\{ \beta_{REm}^i x_{REm}^i \right\}^{\frac{\sigma_m^i - 1}{\sigma_m^i}} + \alpha_{cfm}^i \left\{ \beta_{cfm}^i cf_m^i \right\}^{\frac{\sigma_m^i - 1}{\sigma_m^i}} \right]^{\frac{\sigma_m^i}{\sigma_m^i - 1}} \tag{4b}$$

where:

$z_m^i$, $q_m^i$ = inputting volume of composite intermediate goods and its price;
$x_{REm}^i$, $p_{RE}^i$ = inputting volume of real estate service and price of real estate service;
$cf_m^i$, $pf_m^i$ = inputting volume of composite product factor;
$\tau_m^i$ = net indirect tax rate;
$\alpha_{Zm}^i, \alpha_{REm}^i, \alpha_{cfm}^i, \beta_{Zm}^i, \beta_{REm}^i, \beta_{cfm}^i$ = share parameters ($\alpha_{Zm}^i + \alpha_{REm}^i + \alpha_{cfm}^i = 1$, $\beta_{Zm}^i + \beta_{REm}^i + \beta_{cfm}^i = 1$);
$\gamma_m^i$: scale parameter;
$\sigma_m^i$: elasticity of substitution.

By solving Equation (4), we obtained the demand functions.

$$z_m^i = \frac{1}{\gamma_m^i \left(\beta_{Zm}^i\right)^{1-\sigma_m^i}} \left(\frac{\alpha_{Zm}^i}{q_m^i}\right)^{\sigma_m^i} \Psi_m^{i \frac{\sigma_m^i}{1-\sigma_m^i}} y_m^i \tag{5a}$$

$$x_{REm}^i = \frac{1}{\gamma_m^i \left(\beta_{REm}^i\right)^{1-\sigma_m^i}} \left(\frac{\alpha_{REm}^i}{p_{RE}^i}\right)^{\sigma_m^i} \Psi_m^{i \frac{\sigma_m^i}{1-\sigma_m^i}} y_m^i \tag{5b}$$

$$cf_m^i = \frac{1}{\gamma_m^i \left(\beta_{cfm}^i\right)^{1-\sigma_m^i}} \left(\frac{\alpha_{cfm}^i}{pf_m^i}\right)^{\sigma_m^i} \Psi_m^{i \frac{\sigma_m^i}{1-\sigma_m^i}} y_m^i \tag{5c}$$

where:

$$\Psi_m^i = \left(\alpha_{Zm}^i\right)^{\sigma_m^i} \left(\frac{q_m^i}{\beta_{Zm}^i}\right)^{1-\sigma_m^i} + \left(\alpha_{REm}^i\right)^{\sigma_m^i} \left(\frac{p_{RE}^i}{\beta_{REm}^i}\right)^{1-\sigma_m^i} + \left(\alpha_{cfm}^i\right)^{\sigma_m^i} \left(\frac{\{1 + \tau_m^i\} pf_m^i}{\beta_{cfm}^i}\right)^{1-\sigma_m^i}$$

By substituting Equation (5) into Equation (4a), we obtained the price of goods m in zone *i*.

$$p_m^i = \frac{1}{\gamma_m^i} \Psi_m^{i \frac{1}{1-\sigma_m^i}} \tag{6}$$

The formulations of the next steps are basically the same as the ones above. However, the inputting behavior of product factors is shown below because it has a little difference.

$$pf_m^i cf_m^i = \min_{l_m^i, k_m^i} \left[ w\, l_m^i + r\, k_m^i \right] \tag{7a}$$

$$\text{s.t. } cf_m^i = \gamma_{CFm}^i \left[ \alpha_{Lm}^i \left\{ \beta_{Lm}^i l_m^i \right\}^{\frac{\sigma_{CFm}^i - 1}{\sigma_{CFm}^i}} + \left(1 - \alpha_{Lm}^i\right) \left\{ \left(1 - \beta_{Lm}^i\right) k_m^i \right\}^{\frac{\sigma_{CFm}^i - 1}{\sigma_{CFm}^i}} \right]^{\frac{\sigma_{CFm}^i}{\sigma_{CFm}^i - 1}} \tag{7b}$$

where:

$l_m^i$, $k_m^i$ = inputting volume of labor and capital;
$w, r$ = wage and capital rent;
$\alpha_{Lm}^i$, $\beta_{Lm}^i$ = share parameters;
$\gamma_{CFm}^i$ = scale parameter;
$\sigma_{CFm}^i$ = elasticity of substitution.
By solving Equation (7), we obtained the demand functions.

$$l_m^i = \frac{1}{\gamma_{CFm}^i \left(\beta_{Lm}^i\right)^{1-\sigma_{CFm}^i}} \left(\frac{\alpha_{Lm}^i}{w}\right)^{\sigma_{CFm}^i} \Psi_{CFm}^{i\, \frac{\sigma_{CFm}^i}{1-\sigma_{CFm}^i}} \cdot cf_m^i \tag{8a}$$

$$k_m^i = \frac{1}{\gamma_{CFm}^i \left(1-\beta_{Lm}^i\right)^{1-\sigma_{CFm}^i}} \left(\frac{1-\alpha_{Lm}^i}{r}\right)^{\sigma_{CFm}^i} \Psi_{CFm}^{i\, \frac{\sigma_{CFm}^i}{1-\sigma_{CFm}^i}} \cdot cf_m^i \tag{8b}$$

where:

$$\Psi_{CFm}^i = \left(\alpha_{Lm}^i\right)^{\sigma_{CFm}^i} \left(\frac{w}{\beta_{Lm}^i}\right)^{1-\sigma_{CFm}^i} + \left(1-\alpha_{Lm}^i\right)^{\sigma_{CFm}^i} \left(\frac{r}{1-\beta_{Lm}^i}\right)^{1-\sigma_{CFm}^i}$$

By substituting Equation (8) into Equation (7a), we obtained the price of composite product factors.

$$pf_m^i = \frac{1}{\gamma_{CFm}^i} \Psi_{CFm}^{i\, \frac{1}{1-\sigma_{CFm}^i}} \tag{9}$$

The inputting volume of labor in each zone $i$ is decided by (8a). Therefore, the volume of the employee in zone $i$ is determined, and the households who are employed in zone $i$ decide the residing zone $j$ at the next step.

### 2.3.2. Household's Behavior

We show the outline of locating mechanism of households in this model once again. At first, the volume of firms' products for each zone $i$ is determined in Equation (2), and the inputting volume of labor is yielded from equation (8a), which is decided from the volume of firms' products. The households who are employed in zone $I$ choose the residing zone $j$, based on utility level determined by the accessibility of commuting trips or private trips.

The locating behavior of households that are located in zone $j$ and are working in zone $i$ is shown as a nested structure in Figure 2.

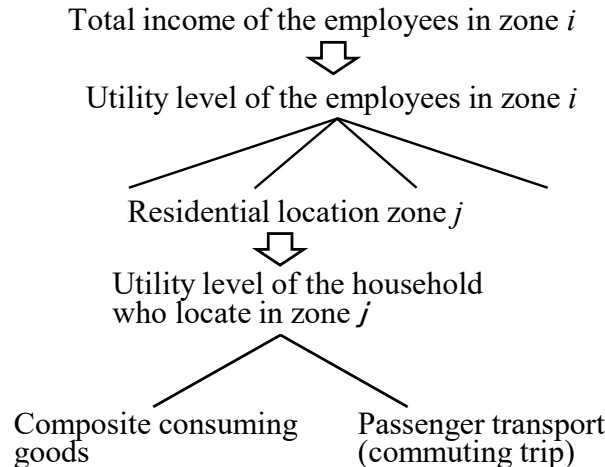

**Figure 2.** Tree structure of household location behavior.

In the first step, the household who works in zone $i$ choose the locating zone where it generates the commuter trips. Its behavior is formulated by an expenditure minimizing program.

$$e_H^i = \min_{u_H^{ij}} \left[ \sum_j p_V^{ij} u_H^{ij} \right] \tag{10a}$$

$$\text{s.t. } u_H^i = \gamma_{LH}^i \left[ \sum_j \alpha_{LH}^{ij} \left\{ \beta_{LH}^{ij} u_H^{ij} \right\}^{\frac{\sigma_{LH}^i - 1}{\sigma_{LH}^i}} \right]^{\frac{\sigma_{LH}^i}{\sigma_{LH}^i - 1}} \tag{10b}$$

where:

$u_H^{ij}$ = utility of household living in zone $j$ and working in zone $i$;

$p_V^{ij}$ = price of utility;

$\alpha_{LH}^{ij}, \beta_{LH}^{ij}$ = share parameters ($\sum_j \alpha_{LH}^{ij} = 1$ , $\sum_j \beta_{LH}^{ij} = 1$);

$\gamma_{LH}^i$ = scale parameter;

$\sigma_{LH}^i$ = elasticity of substitution.

By solving Equation (10), we obtained the utility functions.

$$u_H^{ij} = \frac{1}{\gamma_{LH}^i \left( \beta_{LH}^{ij} \right)^{1-\sigma_{LH}^i}} \left( \frac{\alpha_{LH}^{ij}}{p_V^{ij}} \right)^{\sigma_{LH}^{ij}} \Psi_{LH}^i {}^{\frac{\sigma_{LH}^i}{1-\sigma_{LH}^i}} u_H^i \tag{11}$$

where:

$\Psi_{LH}^i = \sum_n \left( \alpha_{LH}^{ij} \right)^{\sigma_{LH}^i} \left( \frac{p_V^{ij}}{\beta_{LH}^{ij}} \right)^{1-\sigma_{LH}^i}$.

By substituting Equation (11) into Equation (10a), we obtained the expenditure level.

$$e_H^i = \frac{1}{\gamma_{LH}^i} \Psi_{LH}^i {}^{\frac{1}{1-\sigma_{LH}^i}} \cdot u_H^i \tag{12}$$

We assumed that the income for unit labor is constant, and the income of households who work in zone $i$ is obtained as below.

$$\Omega_H^i = [\{wT + rK\}(1 + \tau_H) - S_H] \frac{\sum_m l_m^i}{\sum_j \sum_m l_m^j} \tag{13}$$

where:

$T$ = Total endowment of time in whole region;

$K$ = Total endowment of capital in whole region;

$\tau_H$ = income tax rate;

$S_H$ = saving in whole region.

By substituting Equation (13) into the expenditure level of Equation (12), we obtained the utility level.

$$v_H^i = \frac{\Omega_H^i + r_{RE}^i K_{RE}^i}{p_V^i} \tag{14}$$

where:

$r_{RE}^i$ = land capital rent;

$K_{RE}^i$ = endowment of land capital in zone $i$;

$p_V^i \equiv \frac{1}{\gamma_{LH}^i} \Psi_{LH}^i {}^{\frac{1}{1-\sigma_{LH}^i}}$.

The utility level $u_H^{ij}$ is obtained by substituting $v_H^i$ of Equation (14) into $u_H^i$ of Equation (11). The utility level $u_H^{ij}$ means the one which a household acquires by locating in zone $i$ and determines the volume of consumption. Therefore, we interpret the utility level $u_H^{ij}$ as the location choice of the household.

In the next step, the household decides the input volume of composite goods and passenger transport services on commuting trips. This is formulated by the expenditure minimizing program as below.

$$p_V^{ij} u_H^{ij} = \max_{z_H^{ij}, \, x_{TP_C H}^{ij}} \left[ q_{VH}^j z_{VH}^{ij} + p_{TP}^{ij} x_{TP_C H}^{ij} \right] \tag{15a}$$

$$\text{s.t. } u_H^{ij} = \gamma_{CH}^{ij} \left[ \left(1 - \alpha_{CH}^{ij}\right) \left\{ \left(1 - \beta_{CH}^{ij}\right) z_{VH}^{ij} \right\}^{\frac{\sigma_{CH}^{ij}-1}{\sigma_{CH}^{ij}}} + \alpha_{CH}^{ij} \left\{ \beta_{CH}^{ij} x_{TP_C H}^{ij} \right\}^{\frac{\sigma_{CH}^{ij}-1}{\sigma_{CH}^{ij}}} \right]^{\frac{\sigma_{CH}^{ij}}{\sigma_{CH}^{ij}-1}} \tag{15b}$$

where:

$z_{VH}^{ij}, q_{VH}^j$ = inputting volume of composite goods and its price;

$x_{TP_C H}^{ij}, p_{TP}^{ij}$ = inputting volume of passenger transport services on commuting trips and its price;

$\alpha_{CH}^{ij}, \beta_{CH}^{ij}$ = share parameters;

$\gamma_{CH}^{ij}$ = scale parameter;

$\sigma_{CH}^{ij}$ = elasticity of substitution.

By solving Equation (15), we obtained the demand functions.

$$z_{VH}^{ij} = \frac{1}{\gamma_{CH}^{ij} \left(1 - \beta_{CH}^{ij}\right)^{1-\sigma_{CH}^{ij}}} \left( \frac{1 - \alpha_{CH}^{ij}}{q_{VH}^j} \right)^{\sigma_{CH}^{ij}} \Psi_{CH}^{ij} {}^{\frac{\sigma_{CH}^{ij}}{1-\sigma_{CH}^{ij}}} \cdot u_H^{ij} \tag{16a}$$

$$x_{TP_C H}^{ij} = \frac{1}{\gamma_{CH}^{ij} \left(\beta_{CH}^{ij}\right)^{1-\sigma_{CH}^{ij}}} \left( \frac{\alpha_{CH}^{ij}}{p_{TP}^{ij}} \right)^{\sigma_{CH}^{ij}} \Psi_{CH}^{ij} {}^{\frac{\sigma_{CH}^{ij}}{1-\sigma_{CH}^{ij}}} \cdot u_H^{ij} \tag{16b}$$

where:

$$\Psi_{CH}^{ij} = \left(1 - \alpha_{CH}^{ij}\right)^{\sigma_{CH}^{ij}} \left( \frac{q_{VH}^j}{1 - \beta_{CH}^{ij}} \right)^{1-\sigma_{CH}^{ij}} + \left( \alpha_{CH}^{ij} \right)^{\sigma_{CH}^{ij}} \left( \frac{p_{TP}^{ij}}{\beta_{CH}^{ij}} \right)^{1-\sigma_{CH}^{ij}}$$

By substituting Equation (16) into Equation (15a), we obtain the price of utility.

$$p_V^{ij} = \frac{1}{\gamma_{CH}^{ij}} \Psi_{CH}^{ij}{}^{\frac{1}{1-\sigma_{CH}^{ij}}} \tag{17}$$

Next, the household determines the consuming volume of commodities/services and leisure. In this model, the price $q_{VH}^{j}$ of composite goods in Equation (16a) is not according to zone *i* where the household works, so we built the consuming behavior model of the household for $z_{VH}^{j}$, which is obtained by summing up to working zone *i*.

$$z_{VH}^{j} = \sum_i z_{VH}^{ij} \tag{18}$$

The consuming behaviors of households are shown in Figure 3. In the first step, a household determines the consuming volume of composite goods, real estate service and leisure, respectively. The second step, consuming composite goods, determines the consumption volumes of composite goods, which consist of some goods and freight transport services, and composite services, which consist of some services and passenger transport services. The freight and passenger transport services are assumed to be necessary to consume some composite goods and some services, respectively. In the third step, for consuming the composite of some goods and freight transport services, they decide the inputting volume from some composite goods and freight transport services, respectively, and for some composite goods, it decides the inputting volume of *m* goods. For freight transport services, they choose the origin zone from which freight transport services generate. On the other hand, for the consumption composite of some services and passenger transport services, it chooses the consuming zone and determines the consuming volume of some composite services in each zone. For the consumption volume of services in each zone, it decides the consuming volume of some composite services and passenger transport services, respectively. For the consumption composite of some services, it decides the consuming volume of business, commercial and private services, respectively.

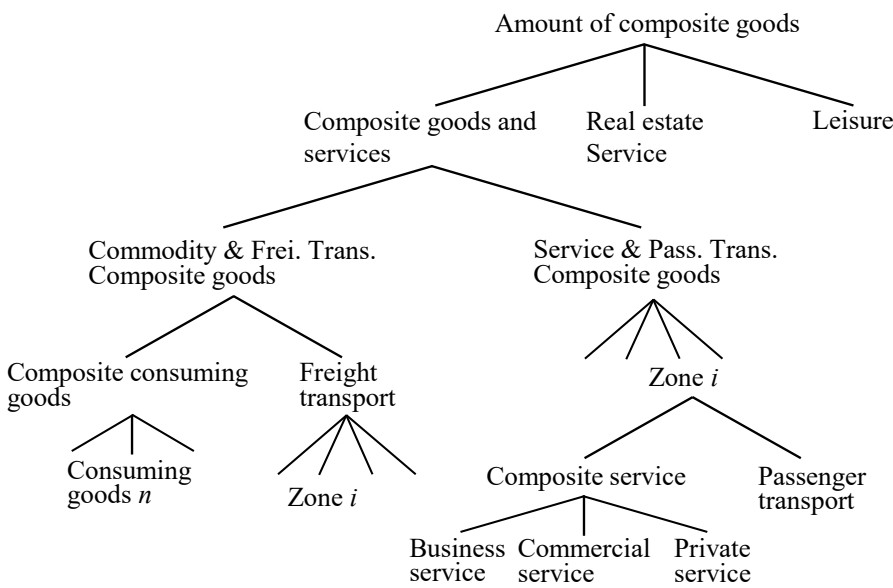

**Figure 3.** Tree structure of household consumption behavior.

These household's consuming behaviors are formulated by the expenditure minimizing program constraint to keep the utility level constant. The formulation of Figure 3 is shown as follows.

$$q_{VH}^j z_{VH}^j = \min_{z_H^j,\, x_{RE\,H}^j, l_H^j} \left[ q_H^j z_H^j + p_{RE}^j x_{RE\,H}^j + w l_H^j \right] \tag{19a}$$

s.t. $z_{VH}^j = \gamma_H^j \left[ \alpha_{ZH}^j \left\{ \beta_{ZH}^j z_H^j \right\}^{\frac{\sigma_H^j - 1}{\sigma_H^j}} + \alpha_{RE\,H}^j \left\{ \beta_{RE\,H}^j x_{RE\,H}^j \right\}^{\frac{\sigma_H^j - 1}{\sigma_H^j}} + \alpha_{LH}^j \left\{ \beta_{LH}^j l_H^j \right\}^{\frac{\sigma_H^j - 1}{\sigma_H^j}} \right]^{\frac{\sigma_H^j}{\sigma_H^j - 1}} \tag{19b}$

where:

$z_H^j$, $q_H^j$ = consuming volume of composite goods and its price;

$x_{RE\,H}^j$, $p_{RE}^j$ = consuming volume of real estate service and price of real estate service;

$l_H^j$, $w$ = consuming volume of leisure and wage;

$\alpha_{ZH}^j, \alpha_{RE\,H}^j, \alpha_{LH}^j, \beta_{ZH}^j, \beta_{RE\,H}^j, \beta_{LH}^j$ = share parameters ($\alpha_{ZH}^i + \alpha_{RE\,H}^i + \alpha_{LH}^i = 1$, $\beta_{ZH}^j + \beta_{RE\,H}^j + \beta_{LH}^j = 1$);

$\gamma_H^j$ = scale parameter;

$\sigma_H^j$ = elasticity of substitution.

By solving Equation (19), we obtained the demand functions.

$$z_H^j = \frac{1}{\gamma_H^j \left( \beta_{ZH}^j \right)^{1-\sigma_H^j}} \left( \frac{\alpha_{ZH}^j}{q_H^j} \right)^{\sigma_H^j} \Psi_H^{j \frac{\sigma_H^j}{1-\sigma_H^j}} \cdot z_{VH}^j \tag{20a}$$

$$x_{RE\,H}^j = \frac{1}{\gamma_m^j \left( \beta_{RE\,H}^j \right)^{1-\sigma_m^j}} \left( \frac{\alpha_{RE\,H}^j}{p_{RE}^j} \right)^{\sigma_m^j} \Psi_m^{i \frac{\sigma_m^j}{1-\sigma_m^j}} \cdot z_{VH}^j \tag{20b}$$

$$l_H^j = \frac{1}{\gamma_H^j \left( \beta_{LH}^j \right)^{1-\sigma_H^j}} \left( \frac{\alpha_{LH}^j}{w} \right)^{\sigma_H^j} \Psi_H^{i \frac{\sigma_H^j}{1-\sigma_H^j}} \cdot z_{VH}^j \tag{20c}$$

where:

$$\Psi_H^j = \left( \alpha_{ZH}^j \right)^{\sigma_H^j} \left( \frac{q_H^j}{\beta_{ZH}^j} \right)^{1-\sigma_H^j} + \left( \alpha_{RE\,H}^j \right)^{\sigma_H^j} \left( \frac{p_{RE}^j}{\beta_{RE\,H}^j} \right)^{1-\sigma_H^j} + \left( \alpha_{LH}^j \right)^{\sigma_H^j} \left( \frac{w}{\beta_{LH}^j} \right)^{1-\sigma_H^j}$$

By substituting Equation (20) into Equation (19a), we obtained the price of composite goods.

$$q_H^j = \frac{1}{\gamma_H^j} \Psi_H^{j \frac{1}{1-\sigma_H^j}} \tag{21}$$

The formulations of the next steps are the same as the ones for firms. Therefore, it is omitted here to show the formulations.

### 2.3.3. Real Estate Firm's Behavior

Real estate firms are also assumed to produce the same behavior as other firms. They also have the tendency of inputting intermediate goods and product factors to produce real estate services. Firms and households secure places to act their economic behaviors by consuming real estate services. It is assumed that the service of an owned house is also supplied by the real estate firm based on the concept of imputed rent.

We assume that firms and households consume the real estate services of the zone they choose to locate, and the real estate firm locates in each zone and provides real estate services to firms and households. If the volume of the location is increased by having a higher quality of life, the inputting volume of real estate service is also increased in its zone. We also assume that the real estate firm produces its service by inputting the land capital of the zone where it supplies its services. When the endowment of land capital in the zone is constant, the land capital rent is raised by increasing the volume of location and inputting more real estate services. Because the real estate service price grows by rising land capital rent, the incentive of firms and households who want to change the location decreases less. Additionally, the location equilibrium has been accomplished as the state of no incentive to change location choice.

We omitted the formulation of real estate firms because it is the same as other firms' formulation (see Section 2.3.1).

2.3.4. Transport Firm's Behavior

In regard to the product of transport service for each OD, it is formulated as below by rearranging Equation (4), which is the behavior model of the firm.

$$p_T^{ki} \, y_T^{ki} = \min_{z_T^{ki}, \, x_{RE\ T}^{ki}, c f_T^{ki}} \left[ q_T^{ki} z_T^{ki} + p_{RE}^{ki} x_{RE\ T}^{ki} + \left(1 + \tau_T^{ki}\right) p f_T^{ki} c f_T^{ki} \right] \tag{22a}$$

$$\text{s.t. } y_T^{ki} = \gamma_T^{ki} \left[ \alpha_{ZT}^{ki} \left\{ \beta_{ZT}^{ki} z_T^{ki} \right\}^{\frac{\sigma_T^{ki}-1}{\sigma_T^{ki}}} + \alpha_{RE\ T}^{ki} \left\{ \beta_{RE\ T}^{ki} x_{RE\ T}^{ki} \right\}^{\frac{\sigma_T^{ki}-1}{\sigma_T^{ki}}} + \alpha_{cfT}^{ki} \left\{ \beta_{cfT}^{ki} c f_T^{ki} \right\}^{\frac{\sigma_T^{ki}-1}{\sigma_T^{ki}}} \right]^{\frac{\sigma_T^{ki}}{\sigma_T^{ki}-1}} \tag{22b}$$

where:
subscript $k,i$ = transport services from zone $k$ to zone $i$;
subscript $T$ = transport firms.

The demand functions by solving Equation (22) are the same as the solutions made previously for other firms. The price of transport service is also similar to the price of services mentioned previously. The same is shown below.

$$p_T^{ki} = \frac{1}{\gamma_T^{ki}} \Psi_T^{ki}^{\frac{1}{1-\sigma_T^{ki}}} \tag{23}$$

where:

$$\Psi_T^{ki} = \left( \alpha_{ZT}^{ki} \right)^{\sigma_T^{ki}} \left( \frac{q_{ZT}^{ki}}{\beta_{ZT}^{ki}} \right)^{1-\sigma_T^{ki}} + \left( 1 - \alpha_{ZT}^{ki} \right)^{\sigma_T^{ki}} \left( \frac{p f_T^{ki}}{1 - \beta_{ZT}^{ki}} \right)^{1-\sigma_T^{ki}}$$

The price of transport service in Equation (23) is yielded for each OD.

Although the transport firm also produces transport service by inputting labor and capital, the inputting efficiency of labor and capital in transport firm is assumed to improve when the required time is reduced by being carried out transport projects. The inputting times of labor and capital are able to be saved by arriving at the destination earlier when the required time is reduced.

Here, we assume that the composite factor function consists of the required time between zones, labor input and capital input and is formulated as homogeneity of degree zero. The composite factor function is formulated as below.

$$\begin{aligned} c f_T^{ki} \left( t_T^{ki}, l_T^{ki}, k_T^{ki} \right) &= c f_T^{ki} \left( \lambda \, t_T^{ki}, \lambda \, l_T^{ki}, \lambda \, k_T^{ki} \right) \\ &= c f_T^{ki} \left( \frac{t_T^{ki\ A}}{t_T^{ki}} t_T^{ki}, \frac{t_T^{ki\ A}}{t_T^{ki}} l_T^{ki}, \frac{t_T^{ki\ A}}{t_T^{ki}} k_T^{ki} \right) \\ &= c f_T^{ki} \left( e f f_T^{ki} \cdot l_T^{ki}, e f f_T^{ki} \cdot k_T^{ki} \right) \end{aligned} \tag{24}$$

where:

$cf_T^{ki}$ = inputting volume of composite factor in transport sector;

$t_T^{ki}$ = required time between zone *j* and *k*;

$l_T^{ki}, k_T^{ki}$ = inputting volume of labor and capital;

$\lambda = \dfrac{t_T^{ki\,A}}{t_T^{ki}} \equiv eff_T^{ki}$;

Subscript *A* = without policy.

The formulation of inputting product factors' behavior in transport firms is shown below.

$$pf_T^{ki} cf_T^{ki} = \min_{l_T^{ki}, k_T^{ki}} \left[ w \cdot l_T^{ki} + r \cdot k_T^{ki} \right] \tag{25a}$$

$$\text{s.t. } cf_T^{ki} = \gamma_T^{ki} \left[ \alpha_{LT}^{ki} \left\{ \beta_{LT}^{ki} eff_T^{ki} \cdot l_T^{ki} \right\}^{\frac{\sigma_T^k - 1}{\sigma_T^{ki}}} + \left( 1 - \alpha_{LT}^{ki} \right) \left\{ \left( 1 - \beta_{LT}^{ki} \right) eff_T^{ki} \cdot k_T^{ki} \right\}^{\frac{\sigma_T^{ki} - 1}{\sigma_T^{ki}}} \right]^{\frac{\sigma_T^{ki}}{\sigma_T^{ki} - 1}} \tag{25b}$$

where:

$\alpha_{LT}^{ki}$, $\beta_{LT}^{ki}$ = share parameters;

$\gamma_T^{ki}$ = scale parameter;

$\sigma_T^{ki}$ = elasticity of substitution.

By solving Equation (25), we obtained the factors' demand functions.

$$l_T^{ki} = \frac{1}{\gamma_T^{ki} \left( \beta_{LT}^{ki} eff_T^{ki} \right)^{1 - \sigma_T^{ki}}} \left( \frac{\alpha_{LT}^{ki}}{w} \right)^{\sigma_T^{ki}} \Psi_T^{ki \frac{\sigma_T^{ki}}{1 - \sigma_T^{ki}}} \cdot cf_T^{ki} \tag{26a}$$

$$k_T^{ki} = \frac{1}{\gamma_T^{ki} \left( \{1 - \beta_{LT}^{ki}\} eff_T^{ki} \right)^{1 - \sigma_T^{ki}}} \left( \frac{1 - \alpha_{LT}^{ki}}{r} \right)^{\sigma_T^{ki}} \Psi_T^{ki \frac{\sigma_T^{ki}}{1 - \sigma_T^{ki}}} \cdot cf_T^{ki} \tag{26b}$$

where:

$$\Psi_T^{ki} = \left( \alpha_{LT}^{ki} \right)^{\sigma_T^{ki}} \left( \frac{w}{\beta_{LT}^k eff_T^{ki}} \right)^{1 - \sigma_T^{ki}} + \left( 1 - \alpha_{LT}^{ki} \right)^{\sigma_T^{ki}} \left( \frac{r}{\{1 - \beta_{LT}^{ki}\} eff_T^{ki}} \right)^{1 - \sigma_T^{ki}}$$

By substituting Equation (26) into Equation (25a), we obtained the price of the composite product factor in transport firms.

$$pf_T^{ki} = \frac{1}{\gamma_T^{ki}} \Psi_T^{ki \frac{1}{1 - \sigma_T^{ki}}} \tag{27}$$

The price of the composite factor is related to the inputting efficiency $eff_T^{ki}$, and the $eff_T^{ki}$ is related to the required time between zones. Thus, when the required time is reduced by transport projects, the price of composite factor goes down through improvement of $eff_T^{ki}$.

### 2.4. Market Equilibrium Conditions

The market equilibrium conditions are shown below in this CGEUE model.

"*n* (Agriculture and Manufacture) goods market"

$$y_n = \sum_i \left( \sum_m x_{n\ m}^i + x_{n\ H}^i \right) + x_{n\ GC} + x_{n\ GI} + x_{n\ I} \tag{28a}$$

"*n* (Services) goods market" $y_n^i = \sum_m x_{nm}^i + x_{nH}^i + x_{nGC}^i + x_{nGI}^i + x_{nI}^i \tag{28b}$

$$\text{"Transport service market" } y_T^{ki} = \sum_m x_{Tm}^{ki} + x_{TH}^{ki} + x_{T\,GC}^{ki} + x_{T\,GI}^{ki} + x_{T\,I}^{ki} \quad (28c)$$

$$\text{"Labor market" } T - \sum_i l_H^i = \sum_i \left( \sum_m l_m^i + l_T^i \right) \quad (28d)$$

"Capital market (except land and building capital)"

$$K = \sum_i \left( \sum_m k_m^i + \sum_T k_T^i \right) \cdots (m: \text{except } RE) \quad (28e)$$

$$\text{"Land and building capital market" } K_{RE}^i = k_{RE}^i \quad (28f)$$

The above market equilibrium conditions show that the markets on agriculture, manufacture goods, labor and the capital, except for land and building capital, are cleared for the whole urban area, the ones on all services except transport services, land and building capital are cleared for each zone, and the ones on transport services are cleared for each OD.

### 2.5. Definition of Benefit

We defined the benefit for some policies based on the concept of equivalent variation (EV). The benefit $ev^{ij}$ is led as below by using the expenditure level of households in Equation (11).

$$ev^{ij} = p_V^{ij\,A} \left( u_H^{ij\,B} - u_H^{ij\,A} \right) \quad (29)$$

where:

subscript *A,B* = means without policy and with policy, respectively.

The incidence benefit of zone *j* $EV^j$ is obtained as follows by summing up with zone *i*, which denotes employing zone of a household.

$$EV^j = \sum_i ev^{ij} \quad (30)$$

## 3. Numerical Results

### 3.1. Outline of Numerical Simulation

Here, we numerically simulate to evaluate urban transport policies for realizing a carbon-free society in the Kofu urban area of Yamanashi Prefecture. Kofu urban area is a typical local city with a population of 534 thousand, and the main transport is automobiles. Therefore, we measured: 1. The policy for converting fossil fuel vehicles to electric vehicles; 2. The policy for improving public transport such as railways or buses; 3. The environmental tax policy; 4. The policy for making the city compact.

Kofu urban area is located in Yamanashi prefecture. In Japan, the input–output table for all prefectures was prepared, and the 2015 Yamanashi input–output table was published. Based on this table, the input–output table of the Kofu urban area was made by proportional division calculation using the household population and each industrial employee population. The industrial sectors in the input–output table are shown in Table 1.

Next, the Kofu urban area was divided into 67 zones, and a transport network was constructed. Figure 4 and Table 2 shows the divided Kofu urban area.

**Table 1.** The industrial sectors in the input–output table.

| | | |
|---|---|---|
| 1 | Agriculture | |
| 2 | Manufacture | |
| 3 | Commerce | |
| 4 | Eating and drinking services | Industrial sector |
| 5 | Public services | |
| 6 | Business and personal services | |
| 7 | Petroleum refinery products | |

**Table 1.** *Cont.*

| | | |
|---|---|---|
| 8 | Electricity | |
| 9 | Water supply | |
| 10 | Real estate services | |
| 11 | Railway transport | |
| 12 | Road passenger transport | |
| 13 | Self-transport (passenger) | Industrial sector |
| 14 | Road freight transport | |
| 15 | Self transport (freight) | |
| 1 | Household | |
| 2 | Government | |
| 3 | Public investment | Final demand sector |
| 4 | Private investment | |

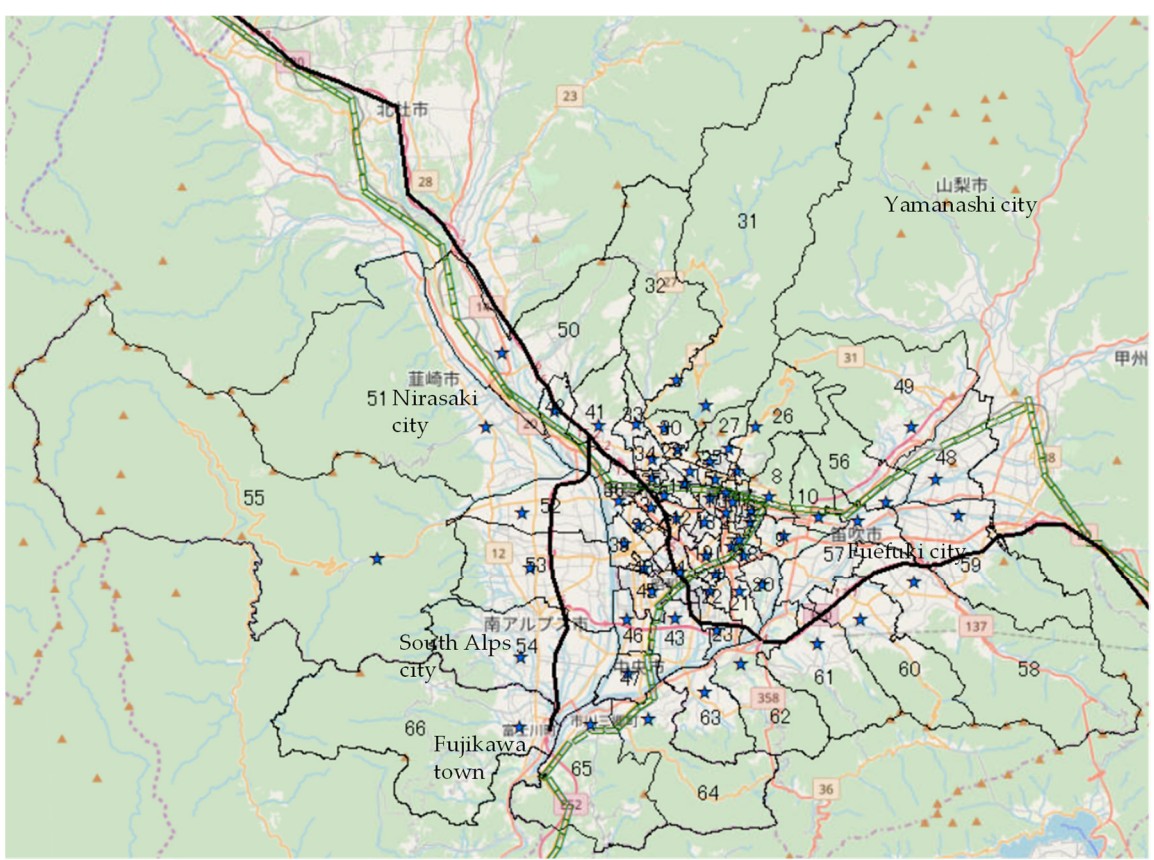

**Figure 4.** The Kofu urban area divided into 67 zones.

**Table 2.** The place name of each 67 zone in Figure 4.

| | | | | | | | | |
|---|---|---|---|---|---|---|---|---|
| 1 | Fujigawa | Kofu city | 24 | Ookuni | Kofu city | 46 | Tatomi North | Chuo city |
| 2 | Aioi | | 25 | Kitashin | | 47 | Tatomi South | |
| 3 | Kasuga | | 26 | Aikawa | | 48 | Yamanashi South | Yamanashi city |
| 4 | Shinkonya | | 27 | Yumura | | 49 | Yamanashi North | |
| 5 | Shiobe | | 28 | Tsukahara | | 50 | Nirasaki east | Nirasaki city |
| 6 | Takumi | | 29 | Chiduka | | 51 | Nirasaki west | |
| 7 | Azuma | | 30 | Haguro | | 52 | Hatta | South Alps city |
| 8 | Satogaki | | 31 | Chiyoda | | 53 | Shirane | |
| 9 | Tamao | | 32 | Shikishima North | Kai city | 54 | Ogasawara | |

**Table 2.** *Cont.*

| | | | | | | |
|---|---|---|---|---|---|---|
| 10 | Kouun | 33 | Shikishima | 55 | Ashiyasu | |
| 11 | Anakiri | | Central-North | 56 | Kasugai | Fuefuki city |
| 12 | Kugawa | 34 | Shikishima Central | 57 | Isawa | |
| 13 | Ishida | 35 | Shikishima South | 58 | Misaka | |
| 14 | Ikeda | 36 | Ryuo | 59 | Ichimiya | |
| 15 | Shinden | 37 | Tomitake Shinden | 60 | Yatsushiro | |
| 16 | Yuda | 38 | Shinohara | 61 | Sakaigawa | |
| 17 | Ise | 39 | Nishiyahata | 62 | Nakamiti | Kofu city |
| 18 | Sumiyoshi | 40 | Tamagawa | 63 | Toyotomi | Chuo city |
| 19 | Kokubo | 41 | Futaba North | 64 | Mitama | Ichikawamisato |
| 20 | Kose | 42 | Futaba West | 65 | Ichikawadaimon | town |
| 21 | Yamashiro | 43 | Tamaho | Chuo city | 66 | Masuho | Fujikawa town |
| 22 | Oosato | 44 | Saijo | Showa town | 67 | Oshikoshi | Showa town |
| 23 | Horinouchi | 45 | Jouei | | | | |

The details of the policies in this study are shown below.

1. The policy for converting fossil fuel vehicles to electric vehicles;

The Japanese Government has set a goal of making 100% electric vehicles for new passenger vehicle sales by 2035 [22]. It is assumed that this political goal will be achieved, and the conversion from fossil fuel vehicles to electric vehicles will progress. As a result of this policy, the running fuel will be converted from petroleum to electric power, so it is expected that GHG emissions will decrease. However, the net price of electric vehicles is higher than that of fossil fuel vehicles because the research and development expenses and new capital investment are required for the development of electric vehicles. Therefore, economic influences generate by its burden.

We evaluate the GHG emissions reduction effect and economic impact of this policy by using the CGEUE model. In other words, we will measure the volume of GHG emissions reduction due to conversion of all fossil fuel vehicles to electric vehicles and economic loss with the equivalent variation (EV). It is assumed that the conversion from fossil fuel vehicles to electric vehicles will reduce the input of petroleum products and instead increase the input of electricity.

2. The policy for improving public transport;

Here, we assume that public transport will be improved so that the average speed of public transport will increase. Specifically, the average railway speed is set from 70 km/h to 75 km/h, and the average bus speed is set from 20 km/h to 30 km/h in the railway and bus network of the Kofu urban area. This will improve the convenience of public transport, and it is expected that a modal shift from automobiles to public transport with less of an environmental burden.

3. The environmental tax policy;

We considered the introduction of an environmental tax, which is thought to be an effective economic method for dealing with environmental issues. The environmental tax shall be levied on fossil fuels of automobiles, and the tax rate is set so that the price of petroleum refinery products will rise by 14.1%. Then, we evaluated the effect of reducing GHG emissions and the deadweight loss due to the introduction of taxes by equivalent variation (EV).

4. The policy for making city compact.

In the policy for making the city compact, we introduced location regulations in the suburb and location deregulations in the city center of the Kofu urban area. In Japan, there is a development permission system for estate developments. By operating this system, when the land and building capitals are rebuilt, it is assumed that the ones in the suburbs are reduced by 1%, and those in the central area are increased by 0.1%. As

a result, the mileage of automobiles can be shortened, and the GHG emissions by being generated by automobiles can be reduced. On the other hand, there is a possibility that an economic burden occurs due to location regulations, so the influence will also be evaluated by equivalent variation (EV).

### 3.2. Results of Policies

#### 3.2.1. Results of the Entire Simulation

The entire numerical results of the five policies are shown in Table 3 and Figure 5, which take into account the combined policy of public transport improvement and environmental taxes in addition to the four policies explained in the previous section. With regard to the combined policy, it showed a numerical result in positive benefit and reduced GHG emission by several numerical calculations.

**Table 3.** The results of the entire simulation.

| | Benefit | Reduction Rate | Change of Real Products (Million Yen/Year) | | | | | | | | | | |
| | (Million Yen/Year) | of GHGs (%) | Railway Trans. | Road Pass. Trans. | Self-Trans. Pass. | Road Frei. Trans. | Self-Trans. Frei. | Agricultue | Manufacture | Commerce | Eat & Drink | Other Services | Real Estate |
|---|---|---|---|---|---|---|---|---|---|---|---|---|---|
| 1.Electric vehicle | −18,630 | −98.92% | 15 | −574 | −10,919 | −9189 | 11,924 | −319 | −16,241 | 1804 | 311 | 9391 | 36 |
| 2.Public transport | 3221 | 0.60% | 23 | 2590 | −7 | 6 | 4 | 6 | 343 | −43 | −21 | 36 | 19 |
| 3.Environmental tax | −19,479 | −6.48% | −3 | −231 | −4656 | −1901 | −2050 | −1321 | −10,937 | 2147 | −1420 | −6319 | −577 |
| 4.City compact | −1 | 0.0016% | 0 | 0 | 1 | 1 | 1 | 0 | 38 | 15 | 4 | 55 | 107 |
| Public Tran. + Tax | 2623 | −1.83% | 25 | 2458 | −1727 | −842 | −758 | 608 | 18,358 | −808 | 251 | −582 | 67 |

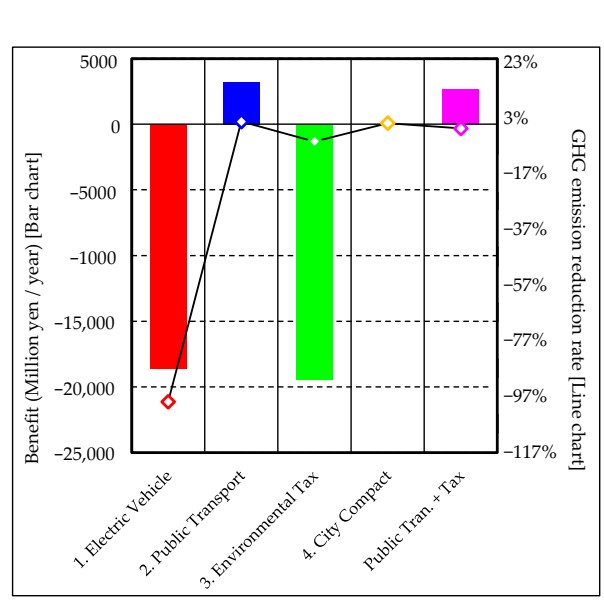

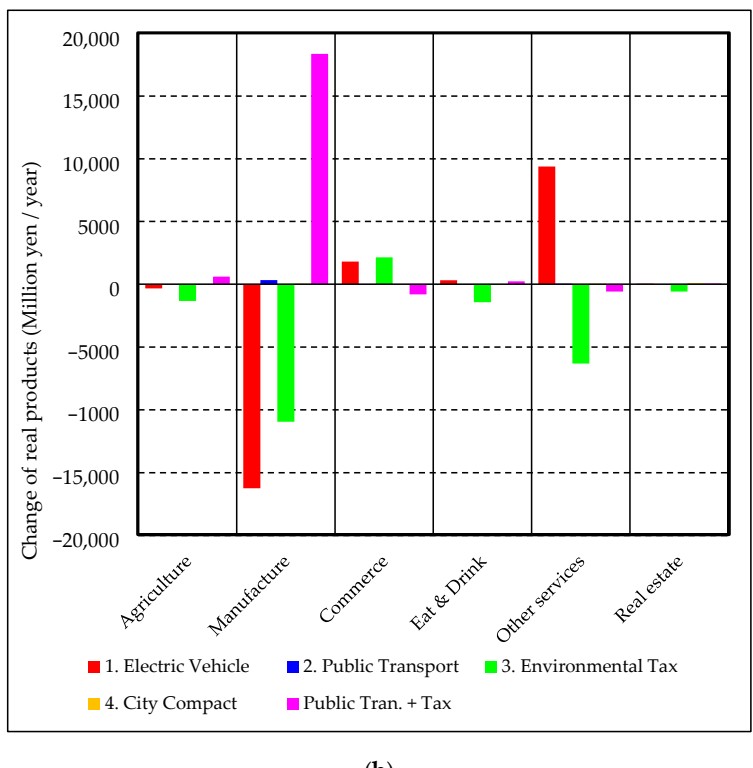

(**a**)

(**b**)

**Figure 5.** The results of the entire simulation: (**a**) the results of benefits and GHG emissions reduction rate; (**b**) the results of real products change for each industrial sector.

These results show that the GHG emissions are greatly reduced, but the benefits are negative under 1. the electric vehicle diffusion policy, where the GHG emissions are increased; the benefits are positive under 2. the public transport improvement policy, where the GHG emissions are reduced; the benefits are negative under 3. the environmental taxes and the GHG emissions are increased; the benefits are also negative although they are minute under 4. making the compact city policy. It seems that the above policies are unlikely to generate positive benefits and reduce GHG emissions. Therefore, we calculated for the integrated policy of 2. public transport improvement policy and 3. environmental taxes to reduce GHG emissions so that positive benefits occurred and GHG emissions were reduced. The results are shown in the bottom column of Table 3.

The following is a discussion of the results of individual policies:

1. The electric vehicle diffusion policy assumes that all fossil fuel vehicles will be converted to electric vehicles, so GHG emissions are significantly reduced. However, the increase in the net price of electric vehicles has reduced the real products of the industrial manufacturing sectors such as transport equipment and also the real products of transport sectors related to automobiles. As a result, it is considered that negative benefits are generated. The results of Table 2. show an increase in the real products in service sectors. This is thought to be due to the demand transformation from manufacturing goods to services;

2. The public transport improvement policy has generated positive benefits. It is probable that the uses of railway and road passenger transport have increased and occurred benefits because the real products of those transport sectors have increased. However, the products of the self-transport passenger sector, which is private automobile transport, decreased slightly. In other words, the increase in the use of railway and road passenger transport was not necessarily due to the conversion from self-transport passengers, but the improvement in public transport generated induced transport, which led to an increase in the use of railway and road passengers;

3. Environmental tax policy reasonably reduces GHG emissions but generates negative benefits. The railway transport sector also inputs road freight transport or self-transport for its own service products, and as a result, the real products of the railway transport sector are declining due to the influence of the environmental tax. The real products of the commerce sector are increasing. This is thought to be transformed demands from other goods to commerce service because the impacts for other goods are relatively larger than the one of commerce services, although the commerce sectors themselves are also affected by the environmental tax;

4. The policy for making the city compact. The land and building capitals in the suburbs were reduced, while they in central areas were increased. As a result, the amount of land and building capital does not change in the whole urban area. As can be seen from the numerical results for each zone later, negative benefits are generated in the suburbs where the land and building capital is regulated, while positive benefits are generated in the central areas where the one has increased. It is thought that the benefits of the whole urban area became almost zero by being canceled out them. GHG emissions are also reduced in the suburbs but increased in the central areas, and those were canceled out. Therefore, the reduction amount became almost zero in the whole urban area. Households that migrated from the suburbs to the central area are actually increasing the number of trips to the suburbs, and as a result, the mileage of automobiles by the households in the central area is increasing.

In the integrated policy, the environmental tax policy reduces the real products in the automobile-related transport sectors. This results in a reduction in GHG emissions. On the other hand, the benefits are not significantly reduced because the real products of the railway and road passenger transport sectors are kept without decreasing.

### 3.2.2. Results of Policies for Each Zone

Next, the results for each zone are shown for each policy.

Figure 6 shows (a) the results of the benefits for each policy by zone and (b) the results of changes in the household population by zone. However, since the changing range is different for each policy, the results are shown separately for the bar chart (right axis) and the line chart (left axis). Figure 7 shows the results of changes in the real products by industry in each zone. Figure 7a–e corresponds to the results of each policy.

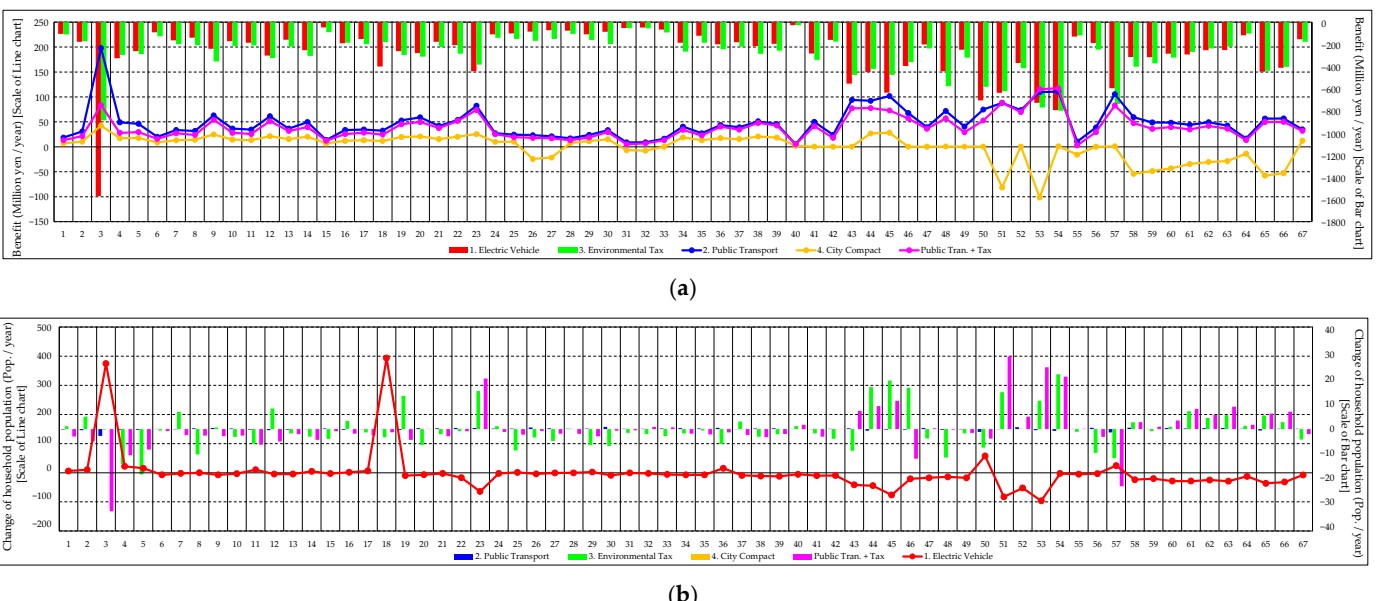

**Figure 6.** The results for each policy by 67 zones: (**a**) the results of benefits for each policy by 67 zones; (**b**) the results of changes in the household population by 67 zones.

From Figure 6a, 1. the electric vehicle diffusion policy and 3. environmental tax policy have negative benefits in all zones; 2. the public transport improvement policy generates a positive benefit in all zones, and 4. the policy for making city compact has a negative benefit in the regulated area of the suburbs and a positive benefit in the central area where land and building capital has increased as described above. Although the integrated policy basically has the same benefit distribution as in 2. the public transport policy, the benefits are slightly less than that due to the influence of the environmental tax.

From Figure 6b, there is almost no change in the household population in 2. the public transport policy and 4. the compact city policy, and the population is increasing in the suburbs in 3. the environmental tax policy. Regarding the change in household population, the integrated policy has almost the same result as 3. the environmental tax policy.

In 1. the electric vehicle diffusion policy, the increase in some household populations is remarkable. The demand for electricity increases sharply due to the diffusion of electric vehicles. Therefore, the products of electricity in the zone where the electricity sector is located will increase (see Figure 7a). It is probable that the increase in products also increases the labor input, and as a result, the household location in the zone increased. Since it is assumed that the total population of households in the Kofu urban area will not change, the population is declining in zones away from the zone where the electricity sector is located.

Figure 7 shows the change in the real products by zone based on the results of Figure 5b. It seems that there is a tendency for the real products to change significantly in the suburbs. However, this is because the size of the zone differs between the central area and the suburbs. Regarding 4. the compact city policy, the change in products of the real estate sector is large because the location changes due to this policy. The tendency is that the real products of the real estate sectors are decreasing in the regulated suburb areas, and it is increasing in the central area.

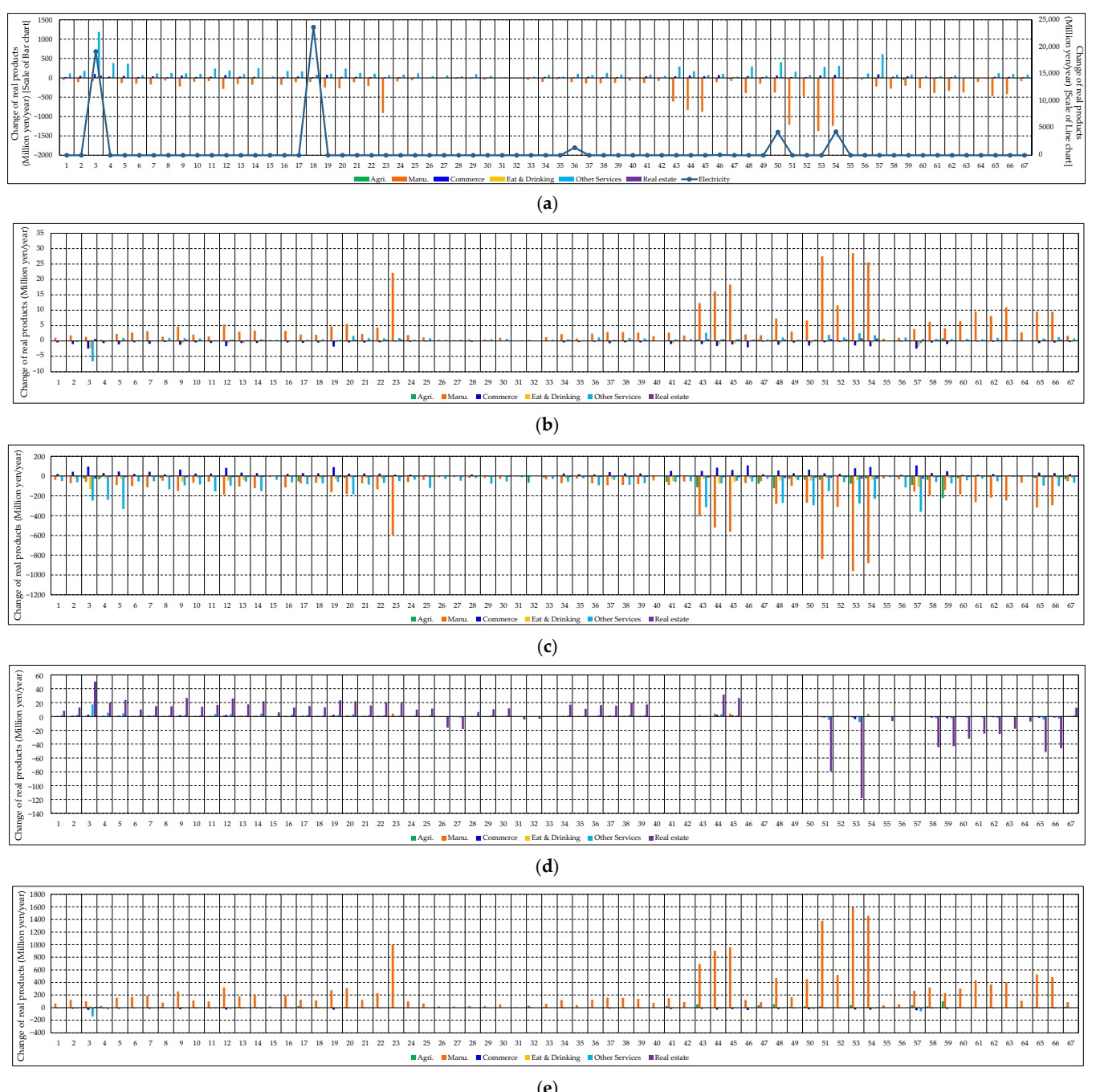

**Figure 7.** The Kofu urban area is divided into 67 zones (**a**–**e**).

## 4. Conclusions

The Government of Japan has declared that it will realize a carbon-free society by 2050. In this study, we examined urban transport policies to realize a carbon-free society in urban areas. In that discussion, we developed the CGEUE model that is integrated the CUE model with the CGE model in order to evaluate not only the GHG emission reduction effect but also the economic impacts of urban transport policies. This CGEUE model is a general equilibrium model that considers the IO structure and deals with both transport and location modeling. In particular, it is characterized by formulating the behavior model of transport firms in detail. It also has the advantage that it can be easily applied to any area where an I–O table can be created. In Japan, an I–O table for each prefecture is created

every five years, and if the one for the target urban area is created, this CGEUE model can be easily applied.

We evaluated policies such as 1. the electric vehicle diffusion policy, 2. the public transport policy, 3. the environmental tax policy and 4. the compact city policy using the CGEUE model. Then, the combination policy of 2. the public transport policy and 4. the compact city policy was evaluated. As a result, we could not find the policies that reduce GHG emissions and generate positive benefits in four policies. Therefore, we were able to propose a policy in which GHG emissions are reduced, and benefits are positive by combining policy 2. and policy 3. Therefore, it can be concluded that the combination policy of policy 2. and policy 3. is the most effective under the condition that it is based on this result.

In this calculation, the diffusion of electric vehicles generates a large GHG emission reduction effect, but the benefits are negative because the net price of electric vehicles is higher than that of fossil fuel vehicles, and economic loss occurs. However, if the technology is further advanced, it may be possible to reduce the fixed costs such as developing electric vehicles and making a capital investment for manufacturing, and it is expected that the electric vehicle price will decrease accordingly. If these are taken into consideration, the effect of the electric vehicles diffusion policy could be measured more accurately.

Regarding 2. the public transport improvement policy, it is assumed here that the required time of public transport, in general, will improve. Since this CGEUE model is considered detail zones, it is also important to find more effective public transport routes, save costs by improving them intensively and propose more effective policies. This is able to change the outcomes of policies that have negative or small benefits here. It is also necessary to verify how the effects and impacts of policies change depending on the structure of each urban area by applying it to other urban areas. In the future, we plan to work on such detailed policy analyses.

**Author Contributions:** Conceptualization, S.M.; methodology, S.M. and A.T.; software, S.M. and H.T.; validation, S.M., H.T. and A.T.; formal analysis, S.M.; investigation, S.M. and H.T.; resources, S.M.; data curation, H.T. and A.T.; writing—original draft preparation, S.M.; writing—review and editing, H.T.; visualization, A.T.; supervision, S.M.; project administration, S.M.; funding acquisition, S.M. All authors have read and agreed to the published version of the manuscript.

**Funding:** This research was funded by JSPS KAKENHI, grant number JP 18K04387, and grant number JP 19K04658.

**Institutional Review Board Statement:** Not applicable.

**Informed Consent Statement:** Not applicable.

**Data Availability Statement:** Not applicable.

**Acknowledgments:** We would like to thank four anonymous reviewers for valuable comments on our paper.

**Conflicts of Interest:** The authors declare no conflict of interest.

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
