# Peer review of "Evaluation of Transport and Location Policies to Realize the Carbon-Free Urban Society"

_sustainability, doi:10.3390/su14010014_

Round 1
Reviewer 1 Report
Review Japan net zero carbon
Aim
The Japanese Government aims for a carbon-free society by 2050. The authors have developed a mathematical model (CGEUE) that is applied to evaluate four policy scenarios in the city of Kofu, Japan.
Method
The starting point is a mathematical simulation model previously developed by the authors and their research colleagues as referenced by half of the entries listed in the bibliography. The model is calibrated with secondary data that relates to the urban area of Koku. Four scenarios are tested in this calibrated model and the results summarised in tabular and graphical form. The results of each policy scenario are described in terms of achieving a carbon-free society.
Evaluation
The first observation is that the English standard is unacceptable for an international journal and the authors should engage a professional editor to revise the manuscript. For example, grammatical errors in the first paragraph: “It has been evaluated…”; On the other hand, it is discussed…”; and “conversing from automobiles to public transport”. Throughout the rest of the manuscript similar errors render the paper hard to understand.
The paper is largely self-referencing without any literature search that determines whether the model adopted is the best way (only way) to evaluate urban policies to realise a carbon-free society.
The substance of the paper is a series of equations that specify the structure of the model (lines 60-347), and presumably these equations appears in earlier publications. Nowhere is it mentioned about some of the universal concepts in modelling: what is the purpose of the model? What is its mathematical structure of the model (yes, see lines 60-347); how are the parameters of this model calibrated?; and how good (or otherwise) is the model when compared with the base-case situation? In particular this paper would be vastly improved if the case study of Kofu included a comprehensive approach to modelling, especially in how the equations of the model were calibrated as currently there is a disjoint between section 2 and section 3.
Minor Comments
There is no reference to the Japanese Government’s policy of a carbon-free society (this reviewer found a METI reference in English but there must be others to cite).
There is no need to include “This is a figure” to all figure captions.
Figure 4 in unreadable.
Figures 6 and 7 containing results for each zone are impossible to interpret given the problem with Figure 4.
Reviewer 2 Report
Finally, I consider that the following comments are taken into account in this regard:
What would be a public policy recommendation to encourage the conversion to electric mobility without a significant increase in the prices of electric vehicles beyond an investment in manufacturing activities?
How would you explain that the development of a compact city does not generate substantial benefits in reducing GHG and economic benefits?
Can this finding in the Japanese context go against global public policies that promote the compact city?
Have you thought that this urban development goes hand in hand with other urban interventions such as the promotion of mixed land uses and local shops, among others?
Reviewer 3 Report
This paper uses a CGEUE model to clarify the impact of the transport and location policies in Kofu city on economic activities at the small zone level. Although the paper attempts a good topic, but there are several gaps in the paper. It needs to be filled before publishing in an international journal.
- Title is appropriate. However, it may be too general. This study focuses on a specific city case.
- In the literature review section in Introduction, it is recommended to analyze other models or approaches and justify why the CGEUE model is useful for than other models.
- In conclusion, discussion part needs to be elaborated. This paper focuses a particular city. It has not been explained whether the results apply to other cities or regions as well. In conclusion, the authors should explain how the results of this paper can be applied to more general cases.
Reviewer 4 Report
Cabon emission is an important topic for not only a specific national but also international context. The topic appears essential. However, I would like to have some comments as follows:
1. The abstract is not clear. The authors do not present the research results and how the results will be applied for policy recommendations.
2. Despite the trade-off between the environmental policies and economic externalities has been raised in the paper. The problem identification is still not sufficient. Specifically, I am not clear what the academic contributions of the paper are? This point makes the introduction part is not strong enough. The reason should be that the literature review is not well conducted. Only several authors are descriptively mentioned rather than analyzed. Furthermore, the viewpoint of the paper is too narrow. It starts from the case of Japan while the contexts of other countries have been neglected.
3. The paper mostly focuses on a technical matter, which is good to generate reliable results. However, the policy discussion is not sufficient. For example, I feel that “the policy for making city compact” is not just simple by saying that land/building density in the suburb or the central area is reduced or increased. Is the policy feasible? How much is the cost of this policy? How should it be reflected in the modeling?
Further, if land use/building density in the suburb is reduced by 1% and in the city center increased by 0.1% (line 395-402), how much the milage of automobiles will be reduced?
Some other comments:
[line 25-26] “The introduction of policies for decarbonization is likely to generate economic impacts while producing environmental improvement effects.” Although I guess “economic impacts” are negative economic impacts. The authors should avoid misleading by making it clearer whether it is negative or positive economic impacts. This issue is evident throughout the paper.
Table 2 and Figure 5 present the same data?
English and writing are not at the standard requirement. I would recommend the paper should be edited by native English speakers before submitting it to the journal.
Round 2
Reviewer 1 Report
None
Reviewer 4 Report
I have no further comments.